# Obstructive Sleep Apnea and Circulating Biomarkers of Oxidative Stress: A Cross-Sectional Study

**DOI:** 10.3390/antiox9060476

**Published:** 2020-06-02

**Authors:** Bernardo U. Peres, AJ Hirsch Allen, Aditi Shah, Nurit Fox, Ismail Laher, Fernanda Almeida, Rachel Jen, Najib Ayas

**Affiliations:** 1Department of Oral Health Sciences, Faculty of Dentistry, University of British Columbia, Vancouver, BC V6T 1Z3, Canada; buperes@mail.ubc.ca (B.U.P.); falmeida@dentistry.ubc.ca (F.A.); 2Department of Medicine, Faculty of Medicine, University of British Columbia, Vancouver, BC V6T 2B5, Canada; ajhirschallen@alumni.ubc.ca (A.H.A.); aditi8@gmail.com (A.S.); Nurit.Fox@vch.ca (N.F.); Rachel.Jen@vch.ca (R.J.); nayas@providencehealth.bc.ca (N.A.); 3Leon Judah Blackmore Sleep Disorders Program, UBC Hospital, Vancouver, BC V6T 2B5, Canada; 4Canadian Sleep and Circadian Network, Montréal, QC H4J 1C5, Canada; 5Department of Anesthesiology, Pharmacology and Therapeutics, Faculty of Medicine, University of British Columbia, Vancouver, BC V6T 2A1, Canada

**Keywords:** obstructive sleep apnea (OSA), biomarkers, oxidative stress

## Abstract

Oxidative stress (OS) drives cardiometabolic diseases. Intermittent hypoxia consistently increases oxidative stress markers. Obstructive sleep apnea (OSA) patients experience intermittent hypoxia and an increased rate of cardiovascular disease, however, the impact of OSA on OS markers is not clear. The objective was to assess relationships between OSA severity and biomarker levels. Patients with suspected OSA referred for a polysomnogram (PSG) provided fasting blood sample. Plasma levels of 8-isoprostane, 8-hydroxydeoxyguanosine (8-OHdG), and superoxide dismutase (SOD) were measured. The relationship between OSA and OS was assessed both before and after controlling for confounders (age, sex, smoking history, history of cardiovascular disease, ethnicity, diabetes, statin usage, body mass index (BMI)). 402 patients were studied (68% male, mean age ± SD = 50.8 ± 11.8 years, apnea-hypopnea index (AHI) = 22.2 ± 21.6 events/hour, BMI = 31.62 ± 6.49 kg/m^2)^. In a multivariable regression, the AHI significantly predicted 8-isoprostane levels (*p* = 0.0008) together with age and statin usage; AHI was not a predictor of 8-OHdG or SOD. Female sex (*p* < 0.0001) and no previous history of cardiovascular disease (*p* = 0.002) were associated with increased antioxidant capacity. Circulating 8-isoprostane levels may be a promising biomarker of the severity of oxidative stress in OSA patients. Prospective studies are needed to determine whether this biomarker is associated with long-term cardiometabolic complications in OSA.

## 1. Introduction

Obstructive sleep apnea (OSA) is the most common respiratory sleep disorder, with an estimated 425 million middle-aged adults worldwide having moderate to severe disease [1]. OSA is characterized by recurrent collapse of the upper airway leading to intermittent hypoxia and sleep fragmentation. Patients with OSA are at increased risk of developing cardiometabolic diseases including cardiovascular disease (CVD), atrial fibrillation, renal disease, hypertension, diabetes, stroke, and metabolic syndrome [2,3,4,5]. 

Oxidative stress is a likely contributor to OSA-related pathologies [6]. Reactive oxygen species (ROS) are chemically reactive molecules produced in the normal metabolism of oxygen. Under stress conditions, ROS levels can increase dramatically, overwhelming antioxidant capacity and therefore leading to a state of oxidative stress (OS). ROS can damage cellular molecules which can cause DNA modification, cell death, apoptosis, and inflammation [7,8,9]. In addition, oxidative stress is recognized increasingly as a fundamental contributor of CVD; indeed, ROS play a role in mediating the adverse effects of many CV risk factors including diabetes, obesity, smoking, and air pollution [7]. Moreover, markers of oxidative stress such as 8-isoprostane levels are an independent risk factor for coronary heart disease [10].

In OSA, analogous to ischemic/reperfusion injury, intermittent hypoxia and consequent reoxygenation can increase the production of ROS [11]. In this regard, rodents exposed to chronic intermittent hypoxia, a validated animal model of OSA, have increased OS biomarkers [12]. Furthermore, studies in OSA patients suggest a potential relationship between many OS biomarkers and OSA, including markers of lipid peroxidation, antioxidant capacity, and DNA oxidation [13]. However, these studies had relatively small numbers of patients (usually <100), lacking the ability to control adequately for confounders, and larger studies are thus needed [14,15].

The objective of this study was to assess the circulating levels of three oxidative stress markers in a large cohort of patients with suspected OSA. Each of these markers measures a different component of OS—specifically, 8-isoprostane is a marker of lipid peroxidation, 8-hydroxydeoxyguanosine (8-OHdG) is a marker of RNA/DNA oxidation, and superoxide dismutase (SOD) is a protective antioxidant enzyme.

## 2. Materials and Methods

### 2.1. Sample and Laboratory Analysis

This study was approved by the University of British Columbia Research Ethics Board (H13-00346) and the Vancouver Coastal Health Research Institutes (V11-80199). Adults (≥19 years old) referred for suspected OSA to the University of British Columbia Hospital sleep disorder laboratory for inpatient polysomnography (PSG) were recruited. Patients were recruited from January 2003 to December 2008. Exclusion criteria included: mentally disabled patients, medically unstable patients, and patients receiving OSA treatment.

PSG was performed using conventional instrumentation and scored according to the recommendations of the American Academy of Sleep Medicine (AASM) [16]. PSG recordings include electroencephalography (EEG) channels, electro-oculograms (left and right), submental electromyograms (EMG), and bilateral tibialis anterior EMG, airflow using nasal pressure and oral thermistor, respiratory efforts using inductance plethysmography belts placed around the chest and abdomen, and oxygen saturation (SaO_2_) with finger pulse oximetry. PSGs were scored by experienced, registered polysomnographic technologists blinded to laboratory results. An obstructive apnea was defined as a decrease in respiratory airflow ≥90% for ≥10 s with continued respiratory efforts; an obstructive hypopnea is defined as a decrease in respiratory airflow of ≥30% for ≥10 s followed by a decrease in SaO_2_ of ≥3% or an arousal. Frequency of apneas and hypopneas was used to calculate the apnea hypopnea index (AHI) per hour of sleep time.

Patients were diagnosed as having OSA based on an AHI of ≥5 events/hour. An AHI between 5 and 15 events/hour was considered mild OSA, 15–30 events/hour was considered moderate, and above 30 events/hour was considered severe.

Consenting patients completed a questionnaire about their family and medical histories, sleep habits and symptoms, mood disorders, alcohol use, smoking status, presence of diabetes (types I and II) and sleepiness on the night of their PSG. History of previous cardiovascular disease (CVD) was determined based on previous diagnosis of hypertension, myocardial infarction, cardiac arrhythmias, angina, and congestive heart failure.

Fasting blood (15 mL) was collected by venipuncture on the morning after PSG, and plasma was stored in a −80 °C freezer. ELISA (Cellbiolabs, CA, USA) was used to test sample levels of 8-isoprostane (Appendix A) and 8 hydroxydeoxyguanosine (8-OHdG). Colorimetric assay (Cellbiolabs, CA, USA) was used to measure superoxide dismutase (SOD) activity. The samples were analyzed in March and April of 2019.

### 2.2. Statistical Analysis

Statistical Analysis Software (SAS version 9.4, SAS Institute Inc., Cary, North Carolina, USA) was used to determine descriptive the statistics on patient demographics/characteristics and levels of oxidative stress markers. We used Pearson`s correlation to assess the relationship between each of the markers and the markers with relevant confounders (continuous variables). To assess the relationship between categorical variables and oxidative stress markers, Student’s t-test was used.

Variables investigated for potential confounding were body mass index (BMI), severity of sleep apnea (AHI), age, sex, smoking status, previous heart disease, diabetes, statins usage, and ethnicity (Caucasians and non-Caucasians). Variables with a *p*-value of less than 0.2 were included in multivariable linear regression models (in addition to age and sex).

## 3. Results

A total of 402 patients were included in the study—baseline characteristics are shown in Table 1. Most of the patients were Caucasian and the majority were male (68%), with a mean age of 50.8 years. The frequency distribution of mild (30%), moderate (26%), and severe OSA (25%) was similar in the cohort. In general, the patients had moderate OSA (mean AHI was 22.2/h) and were obese (mean BMI 31.6 kg/m^2^).

Increased BMI tended to be associated with increased levels (*p* = 0.07) while statin use tended to decrease levels (*p* = 0.17). 8-OhDG levels were only associated with sex (*p* = 0.017), with a greater level in males. Increased SOD activity was significantly associated with increasing BMI and female sex, with a trend for increasing AHI (*p* = 0.08); presence of heart disease was associated with a decreased level (*p* = 0.09) (Table 2). A significant negative relationship was found between 8-isoprostane and SOD activity (*r* = −0.31, *p* < 0.0001) but not between 8-isoprostane and 8-OhDG, or between SOD and 8-OhDG (*p* > 0.2) (Table 2).

Based on a threshold *p*-value < 0.2, we constructed multivariable models including these variables, age and sex (Table 3). Significant independent predictors of increased 8-isoprostane levels included increasing AHI, reduced age, and non-use of statins. AHI was not an independent predictor of the other two markers. SOD activity was significantly increased in females, while BMI and the presence of heart disease decreased levels. Percent below 90% of oxygen saturation was initially included in the model for isoprostane, however, due to the high correlation with AHI and no additional effect on the model we excluded this from the final model (although when it was included with AHI, AHI still remained an independent predictor (*p* = 0.0013) while saturation was not).

## 4. Discussion

We investigated markers of lipid peroxidation (8-isoprostane), DNA degradation (8-OHdG), and antioxidant capacity (SOD) in a large cohort of patients with suspected OSA. OSA severity was independently associated with circulating 8-isoprostane levels even after controlling for relevant confounders such as age, sex, BMI, and statin usage. However, AHI was not independently associated with the other biomarkers of OS that we measured. Female sex, and no previous history of cardiovascular disease were associated with increased SOD activity.

Animal studies consistently demonstrate that intermittent hypoxia increases a broad range of OS biomarkers [17]. For example we have shown that exposure of mice to 8 weeks of intermittent hypoxia (fraction of FiO_2_ reduced to 6%, 60/h during the day for 8 weeks) increased circulating levels of isoprostane [6], while another showed increased levels 8-OHdG levels with intermittent hypoxia exposure [18]. Our results differ somewhat from these animal investigations. In our study of patients with OSA, only 8-isoprostane levels were associated with AHI while the other biomarkers were not. These differences might be related to several factors including differences in species, duration of hypoxia (weeks vs. years), and the intensities of hypoxia stimuli; the degree of hypoxia is more severe in most animal models than in human OSA. For example, an exposure to 6% FiO_2_ for 30 s every minute results in an oxy-hemoglobin saturation of 55–60% [6]. In contrast, the degree of desaturation in patients with OSA tends to be more modest: in our cohort, even patients with severe OSA only spent 12.29% of the study below 90% on average.

Our study confirms and extends the results of previous investigations in terms of an association between 8-isoprostane levels and OSA. Carpagnano and colleagues reported that 8-isoprostane breath condensation levels in 18 patients with OSA were elevated compared to obese subjects and health controls [19]. A meta-analysis of 222 patients with OSA and 194 controls showed substantial increases in the standardized mean difference (*g* = 1.1) of 8-isoprostane levels [15]. However, this study combined different sources of 8-isoprostane (plasma, exhaled breath condensate, and urine) and the results were not adjusted for known confounding factors. Our study exceeds the sample size of all studies included in this recent meta-analysis (222 patients with OSA versus 329) and is controlled for important confounders.

In our study, severity of OSA was not associated with 8-OHdG. However, a study by Pialoux and colleagues reported increases of 40–46% (*p* < 0.05) in 8-OHdG when subjects were exposed to an intermittent hypoxia protocol [20]. This study only included 10 healthy male subjects, and they were exposed to a fairly significant degree of oxygen desaturation over a short time (4 days) only during the daytime while awake. The applicability of these findings to humans with OSA is thus unclear.

Our study is particularly relevant given the association between 8-isoprostane and cardiovascular disease in non-OSA cohorts [10]. In women, high urinary levels are associated with doubled odds of developing cardiovascular disease [21]. 8-isoprostane is associated with many risk factors for coronary heart disease including obesity and smoking [10]. Also, accepted markers of cardiovascular disease, such as c-reactive protein (CRP) [22], correlate with 8-isoprostane (*r* = 0.097, *p* < 0.001) [23].

The demonstration of an independent association between OSA severity and isoprostane in our study, together with the animal work showing increased levels of isoprostane with experimentally induced intermittent hypoxia [6] suggests that OSA is a cause of OS. In turn, this OS most likely partially drives the greater rates of cardiometabolic complications seen in OSA patients. Although prospective studies need to be done using robust clinical outcomes to verify this, isoprostane might represent a useful biomarker of cardiovascular risk in patients with OSA and could be used to help stratify risk at the time of diagnosis. This is important as currently there is a paucity of studies published using prospective biomarkers in the context of OSA [24]. Furthermore, this biomarker might represent a reasonable intermediate target for reduction by therapies for OSA (e.g., continuous positive airway pressure (CPAP) ) or antioxidants.

There are differences between men and women in the oxidative stress response to stimulation by intermittent hypoxia, which is believed to be related to estrogen variations governing ROS production [25]. Our study indicates that sex was independently associated with SOD levels. More specifically, more than half of the subjects in the highest quartile of SOD levels were females (data not shown), suggesting that women in our study cohort had increased antioxidant capacity. Our adjusted (Table 3) and unadjusted analyses (Table 2) indicate the significance between female sex and SOD levels. A similar sex difference was also reported by Wang et al. [26] in their multivariate regression analysis of SOD levels and documented coronary artery disease in 590 patients. In contrast to our findings, they found that age and smoking status were also associated with SOD levels, while our results indicated that the relationship between age and SOD was not statistically significant (*p* = 0.079), as was the relationship between smoking status and SOD levels (*p* = 0.81). It is essential to highlight that in contrast to our study, the study by Wang et al. did not adjust for sleep parameters.

There are many strengths to our study. First, we measured a broad range of OS stress markers in a large cohort. Second, sleep parameters were obtained from inpatient PSG as opposed to questionnaires or home-sleep studies. Third, we were able to control for a number of important confounders including BMI. Fourth, we used circulating levels of markers drawn at the same time of day (morning fasting). However, we also acknowledge a number of limitations to our study. First, the history of cardiovascular disease and smoking was based on self-reported data as opposed to medical-chart review. Second, the study was done in one center in Canada—these results may not be generalizable outside of this population. Third, the degree of desaturation was fairly modest in our study (i.e., on average 4.62% of the study below 90%). It is possible that there may have been a greater impact on biomarkers such as SOD and 8-OHdG if patients with more substantial desaturation were studied. Fourth, we acknowledge that there was a long time-period between collection of samples and processing of samples. We cannot rule out the possibility that this might have affected accuracy of the tests. However, we think this would be unlikely as factors other than storage time in a deep freezer are generally considered more important, including repeated freeze/thaw episodes and differences in immuno-assay activity over time. We did not freeze/thaw any of the samples, and completed all the measurements of the oxidative stress markers using the same assay to reduce variability. Also, there is little data suggesting change of activity over time [27]. For example, in one study, there was no correlation between circulating 8-isoprostane levels and storage time in samples stored for up to 7 years [28]. In addition, many other studies have successfully used frozen samples stored for many years, including a recent one using multiple Luminex assays to test 85 circulating proteins in samples collected from 1998 to 2005 [29], and a technical review showed stability of the markers investigated in our paper of months to years [30].

## 5. Conclusions

OSA severity was an independent predictor of circulating 8-isoprostane levels, a marker of lipid oxidation, in a clinic-based cohort. However, OSA severity was not associated with antioxidant and DNA oxidation markers. 8-isoprostane may thus represent a useful biomarker of oxidative stress in OSA. Future prospective studies should focus on determining whether this marker is predictive of adverse long-term health outcomes such as cardiometabolic disease, and whether levels can be affected by therapies for OSA (e.g., CPAP) or other therapies (e.g., antioxidants).

## Figures and Tables

**Table 1 antioxidants-09-00476-t001:** Baseline characteristics.

Patient Profile	Entire Cohort (*n* = 402)	No OSA (*n* = 71)	Mild OSA (*n* = 123)	Moderate OSA (*n* = 105)	Severe OSA (*n* = 103)
Age (years)	50.86 ± 11.84	45.28 ± 11.40	51.29 ± 11.54	51.68 ± 10.96	53.43 ± 12.27
Males (%)	68.40	53.42	61.79	71.43	83.65
BMI (kg/m^2^)	31.62 ± 6.49	30.36 ± 6.63	30.91 ± 6.57	32.29 ± 6.51	32.68 ± 6.12
AHI (events/h)	22.24 ± 21.61	2.33 ± 1.61	9.32 ± 2.79	21.55 ± 3.99	51.91 ± 20.67
% Time Below 90% SaO_2_	4.62 ± 12.09	0.6 ± 2.9	1.53 ± 4.47	3.30 ± 8.21	12.29 ± 19.73
ESS	7.42 ± 6.15	7.19 ± 6.29	7.37 ± 5.91	7.88 ± 6.20	7.21 ± 6.35
Heart Disease (%) *	22.85	13.70	27.64	22.86	23.81
Smokers (%)	7.37	4.11	7.32	6.67	10.48
Diabetes (%)	8.85	6.85	8.13	7.62	12.38
Caucasians (%)	80.51	84.06	82.50	78.85	77.23
Statins Users (%)	20.15	12.33	18.70	20.95	26.67
**Oxidative Stress Markers**					
8-isoprostane (pg/mL)	3186.48 ± 3922.69	2395.08 ± 1708.61	2725.35 ± 3013.37	3090.36 ± 3160.93	4422.97 ± 5983.41
8-OHdG (ng/mL)	15.92 ± 7.46	15.28 ± 8.46	16.40 ± 7.20	16.65 ± 7.33	15.25 ± 7.23
SOD (%)	18.29 ± 9.45	19.74 ± 10.05	18.22 ± 9.51	19.41 ± 9.02	16.28 ± 9.2

AHI: apnea-hypopnea index; BMI: body mass index; ESS: Epworth sleep scale; OHdG: hydroxydeoxyguanosine; SOD: superoxide dismutase. ******* Heart disease included hypertension, myocardial infarction; cardiac arrhythmias, angina, and congestive heart failure.

**Table 2 antioxidants-09-00476-t002:** Univariate Analysis.

Variable	8–Isoprostane (pg/mL)		8-OhdG (ng/mL)		SOD Activity (%)	
Pearson’s Correlation	r	*p*-value	r	*p*-value	r	*p*-value
**Age**	0.1114	0.0257	0.0423	0.3953	0.0030	0.9511
**BMI**	0.0918	0.0661	0.0583	0.2406	0.1133	0.0226
**AHI**	0.1646	0.0008	0.005	0.9157	0.0866	0.0819
**% Time Below 90% SaO_2_**	0.0761	0.1280	0.003	0.9367	0.0395	0.4283
Student’s T-test	*t*-value	*p*-value	*t*-value	*p*-value	*t*-value	*p*-value
**Sex ***	0.90	0.3672	2.40	0.0173	4.42	<0.0001
**Statin Usage**	1.40	0.1656	0.06	0.9484	0.74	0.4631
**Diabetes**	0.12	0.9076	1.04	0.3021	0.56	0.5807
**Ethnicity ****	0.76	0.5814	0.77	0.5723	0.86	0.5074
**Heart Disease *****	0.49	0.6263	1.01	0.3164	1.71	0.0897
**Smoking Status**	0.83	0.4096	0.13	0.8936	0.00	0.9981

AHI: apnea-hypopnea index; BMI: body mass index; ESS: Epworth sleep scale; OHdG: hydroxydeoxyguanosine; SOD: superoxide dismutase. * Sex: reference males; ** ethnicity divided into Caucasian and non-Caucasian; *** heart disease included hypertension, myocardial infarction, cardiac arrhythmias, angina, and congestive heart failure.

**Table 3 antioxidants-09-00476-t003:** Multiple Linear Regression Models.

Variable	8–Isoprostane (pg/mL)			8-OhdG (ng/mL)			SOD Activity (%)		
	Estimate	SE	*p*-Value	Estimate	SE	*p*-Value	Estimate	SE	*p*-Value
**Age**	60.32	17.49	0.0006	0.02704	0.031	0.3870	0.0470	0.0409	0.2512
**Sex (female)**	132.27	431.21	0.7592	1.8299	0.794	0.0217	4.02	1.0244	<0.0001
**BMI**	37.74	31.98	0.2388	-	-	-	0.1503	0.0736	0.0420
**AHI**	32.13	9.492	0.0008	-	-	-	−0.034	0.022	0.1235
**Statin Usage**	−1172.65	515.17	0.0234	-	-	-	-	-	-
**Heart Disease ***	-	-	-	-	-	-	−2.6147	1.14	0.0229

**Abbreviations**: AHI: apnea-hypopnea index; BMI: body mass index; ESS: Epworth sleep scale; OHdG: hydroxydeoxyguanosine; SOD: superoxide dismutase. ******* Heart disease included hypertension, myocardial infarction, cardiac arrhythmias, angina, and congestive heart failure—variables not included.

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
