# Peer review of "Obstructive Sleep Apnea and Circulating Biomarkers of Oxidative Stress: A Cross-Sectional Study"

_antioxidants, 2020, doi:10.3390/antiox9060476_

Round 1
Reviewer 1 Report
Patients with Obstructive sleep apnea (OSA) present an increase risk of cardiovascular diseases and others metabolic syndromes. In this manuscript entitled "Obstructive sleep apnea and circulating biomarkers of oxidative stress: A Cross-sectional study” by B.U. Peres and co-authors, the association between OSA-induced hypoxia and oxidative stress level was investigated in a large cohort of OSA patients (n=402), by analysis of three plasma biomarkers levels: 8-isoprostane, 8-hydroxydeoxyguanosine (8-OHdG) and superoxide dismutase (SOD). Interestingly, a large set of cofounders was included in the analysis to address the significance of the biomarkers and possible correlations. Circulating 8-isoprostane appeared as the only marker for OSA severity independently of others confounders such as age, sex, BMI, and statin usage. Another interesting correlation indicated a significant association between increased SOD activity and female sex.
The project is nicely introduced, positioning the question in a relevant context. Methods section provides details about the procedures and statistical analysis. However, The blood samples were collected over a total period of five years (2003-2008) and samples were kept frozen until March-April 2019, when the sera analyses took place, which raise questions about the accuracy of the tests. In particular what is the test used to assess for SOD activity: is it a measure of the protein level by Elisa or western-blot, or a direct measure of the enzymatic activity of the enzyme? The value indicated in Table 2 is activity (%). What is the reference? These information and the data obtained need to be provided (possibly as supp. Information).
A second minor revision concerns the weird sentence about Supplementary Materials, which needs to be removed I guess.
In conclusion, analysis of this physiological cohort position 8-isoprostane, a marker of lipid peroxidation, as a relevant marker for OSA severity, and open avenues for the correlation with other possible related cardiometabolic complications.
Author Response
May 27, 2020
To the Editor,
Thank you for reviewing our manuscript and considering it for potential publication. We have tried to address the helpful comments of the reviewer 1 in a point by point fashion and believe the manuscript is much stronger as a result.
Reviewer 1
Patients with Obstructive sleep apnea (OSA) present an increase risk of cardiovascular diseases and others metabolic syndromes. In this manuscript entitled "Obstructive sleep apnea and circulating biomarkers of oxidative stress: A Cross-sectional study” by B.U. Peres and co-authors, the association between OSA-induced hypoxia and oxidative stress level was investigated in a large cohort of OSA patients (n=402), by analysis of three plasma biomarkers levels: 8-isoprostane, 8-hydroxydeoxyguanosine (8-OHdG) and superoxide dismutase (SOD). Interestingly, a large set of cofounders was included in the analysis to address the significance of the biomarkers and possible correlations. Circulating 8-isoprostane appeared as the only marker for OSA severity independently of others confounders such as age, sex, BMI, and statin usage. Another interesting correlation indicated a significant association between increased SOD activity and female sex.The project is nicely introduced, positioning the question in a relevant context. Methods section provides details about the procedures and statistical analysis.
Thank you for your kind comments.
However, the blood samples were collected over a total period of five years (2003-2008) and samples were kept frozen until March-April 2019, when the sera analyses took place, which raise questions about the accuracy of the tests.
This was a concern for us as well. However, we believe our results are accurate for a number of reasons. First, factors other than storage time in a deep freezer are generally considered more important, including repeated freeze/thaw episodes and differences in immuno-assay activity over time. We did not freeze/thaw any of the samples, and all the measurements of the oxidative stress markers were done in batch, using the same assay to reduce variability. It must also be noted that samples were centrifuged and frozen quickly (within 30 mins) after obtaining them. Second, there is little data suggesting change of activity over time (Arts, E.E. et al. Serum Samples That Have Been Stored Long-Term (>10 Years) Can Be Used as a Suitable Data Source for Developing Cardiovascular Risk Prediction Models in Large Observational Rheumatoid Arthritis Cohorts. BioMed Res. Int. 2014). For example, in one study, there was no correlation between circulating 8-isoprostane levels and storage time in samples stored for up to 7 years (Cracowski JL. Isoprostanes as a tool to investigate oxidative stress in scleroderma spectrum disorders--advantages and limitations. Rheumatology (Oxford) 2006; 45: 922-923). In addition, many other studies have successfully used frozen samples stored for many years, including a recent one using multiple Luminex assays to test 85 circulating proteins in samples collected 1998-2005 (Ho JE et al. Am Heart Assoc. 2018), and a technical review technical reviews showed stability of the markers investigated in our paper of months to years (Frijhoff, J. et al. Clinical Relevance of Biomarkers of Oxidative Stress. Antioxid. Redox Signal. 2015).
The following has been added to the Discussion Section (Limitations) with appropriate references:
“Fourth, we acknowledge that there was a long time period between collection of samples and processing of samples. We cannot rule out the possibility that this might have affected accuracy of the tests. However, we think this would be unlikely as factors other than storage time in a deep freezer are generally considered more important, including repeated freeze/thaw episodes and differences in immuno-assay activity over time. We did not freeze/thaw any of the samples, and all the measurements of the oxidative stress markers in batch, using the same assay to reduce variability. Also, there is little data suggesting change of activity over time. For example, in one study, there was no correlation between circulating 8-isoprostane levels and storage time in samples stored for up to 7 years. In addition, many other studies have successfully used frozen samples stored for many years, including a recent one using multiple Luminex assays to test 85 circulating proteins in samples collected 1998-2005, and a technical review showed stability of the markers investigated in our paper of months to years.”
In particular what is the test used to assess for SOD activity: is it a measure of the protein level by Elisa or western-blot, or a direct measure of the enzymatic activity of the enzyme? The value indicated in Table 2 is activity (%). What is the reference? These information and the data obtained need to be provided (possibly as supp. Information).
Thank you for this comment. We used the Superoxide Dismutase Activity Assay from CellBiolabs, it is a colorimetric assay. It uses xanthine/xanthine oxidase system to generate superoxide anions. These anions generated by the system are detected by a chromagen solution; however in the presence of SOD the concentration of anions is smaller, reducing the color seen in the wells. The activity of SOD is determined by the inhibition of chromagen reduction. We have modified the materials and methods section to specify that SOD was measured using a colorimetric assay. Lines 88-91 of the manuscript were edited to the following: “ELISA and colorimetric assays (Cellbiolabs, CA, USA) were used to test sample levels of 8-isoprostane, 8-hydroxydeoxyguanosine (8-OHdG) and superoxide dismutase (SOD). The samples were analyzed in March and April of 2019. To ELISA (Cellbiolabs, CA, USA) was used to test sample levels of 8-isoprostane and 8 hydroxydeoxyguanosine (8-OHdG). Colorimetric assay (Cellbiolabs, CA, USA) was used to measure superoxide dismutase (SOD) activity. The samples were analyzed in March and April of 2019.” Finally, a supplemental material will be provided to include the information above and more details about the data (including comments from Reviewer 1 and 2).
A second minor revision concerns the weird sentence about Supplementary Materials, which needs to be removed I guess.In conclusion, analysis of this physiological cohort position 8-isoprostane, a marker of lipid peroxidation, as a relevant marker for OSA severity, and open avenues for the correlation with other possible related cardiometabolic complications.
We apologize for this error. Of note, the edited version of the manuscript will now have a supplemental table, and this will be edited accordingly.
Sincerely yours,
Bernardo Peres, Ismail Laher, and Najib Ayas on behalf of the authors
Reviewer 2 Report
The scientific manuscript by Peres et al. investigates the association between obstructive sleep apnea and three oxidative stress biomarkers: 8-isoprostane for lipid peroxidation, 8-OHdG for nucleic acid oxidation, and SOD a an antioxidant. Study is well designed and results clearly presented. Authors data indicates that 8-isoprostane has potential to be used as a biomarker to assess the severity of oxidative stress in patients with obstructive sleep apnea. Below some comments to improve clarity on data presentation.
- In Line 105, authors wrote that in general, patients had moderate OSA. However the table show a similar frequency distribution between mild, moderate, and severe OSA. Please rephrase for accuracy.
- Table 1 suggests that the frequency of smokers and diabetes is higher in severe OSA than other groups. Was this assessed? Is this significant? Please clarify.
- Measurement values of 8-isoprostane suggest a technical artifact. Table shows that the SD/SE values are higher than the mean. Since measurement values cannot be negative, this highlights a technical issue. What was the technical limit for minimal values accurately assesed for 8-isoprostane? Please revise and show the standard curve used for quality control. Present a table with frequency of non-detectable values.
- In Line 108-109, authors wrote that increased AHI and decreased age were significantly associated with elevated 8-isoprostane levels. While the univariate analysis demonstrates this is statistically significant, the r correlation value is 0.16 and 0.11 respectively which indicates a very minor/weak association. Please rephrase for accuracy.
Author Response
May 27, 2020
To the Editor,
Thank you for reviewing our manuscript and considering it for potential publication. We have tried to address the helpful comments of the reviewer 2 in a point by point fashion and believe the manuscript is much stronger as a result.
Reviewer 2
The scientific manuscript by Peres et al. investigates the association between obstructive sleep apnea and three oxidative stress biomarkers: 8-isoprostane for lipid peroxidation, 8-OHdG for nucleic acid oxidation, and SOD an antioxidant. Study is well designed and results clearly presented. Authors data indicates that 8-isoprostane has potential to be used as a biomarker to assess the severity of oxidative stress in patients with obstructive sleep apnea.
Thank you for your kind comments about our manuscript.
Below some comments to improve clarity on data presentation.
- In Line 105, authors wrote that in general, patients had moderate OSA. However the table show a similar frequency distribution between mild, moderate, and severe OSA. Please rephrase for accuracy.
We apologize for the confusion. We rephrased line 105 as suggested. Original : In general, they had moderate OSA (mean AHI was 22.2/hr) and were obese (mean BMI 31.6 kg/m2). Modified: “The frequency distribution of mild (30%), moderate (26%) and severe OSA(25%) was similar in the cohort. On average, they had a moderate degree OSA (mean AHI was 22.2/hr) and were obese (mean BMI 31.6 kg/m2).”
- Table 1 suggests that the frequency of smokers and diabetes is higher in severe OSA than other groups. Was this assessed? Is this significant? Please clarify.
The purpose of Table 1 was to describe the cohort of patients, and levels of significance in different variables are usually not tested. The idea was to provide a snapshot of the people being tested and what are the differences seen in different subgroups. These variables were taken into consideration when adjusting for the tests of interest in the manuscript. Smoking Status and Diabetes were not significantly associated (p>0.05) with levels of oxidative stress markers (Table 2).
- Measurement values of 8-isoprostane suggest a technical artifact. Table shows that the SD/SE values are higher than the mean. Since measurement values cannot be negative, this highlights a technical issue. What was the technical limit for minimal values accurately assesed for 8-isoprostane? Please revise and show the standard curve used for quality control. Present a table with frequency of non-detectable values.
Indeed, there is a high variation on the values of 8-isoprostane, providing a standard deviation higher than the mean. All samples were done in duplicate and fitted to their own plate’s standard curve. Each plate had 40 samples. The total of samples outside the detected range was 2% (11 out of 402). Of those, 8 presented with a value higher than the upper limit of the standard curve, and 3 lower. The following table will be provided as supplemental material:
Table S 1 – Technical Data on 8-isoprostane Assays
|
Plate |
Curve Fit R2 |
CV of Standard Curve (%) |
CV of Samples (%) |
Samples Above Detectable Range (count) |
Samples Below Detectable Range (count) |
|
Pilot |
1.000 |
4.928 |
18.28 |
0 |
0 |
|
1 |
0.999 |
3.816 |
15.81 |
0 |
0 |
|
2 |
0.999 |
4.071 |
33.85 |
0 |
0 |
|
3 |
0.999 |
3.014 |
17.31 |
4 |
1 |
|
4 |
0.999 |
7.471 |
17.49 |
0 |
0 |
|
5 |
0.999 |
4.551 |
10.81 |
0 |
0 |
|
6 |
0.999 |
5.901 |
18.73 |
2 |
1 |
|
7 |
0.999 |
3.916 |
14.7 |
2 |
1 |
|
8 |
0.998 |
10.071 |
13.19 |
0 |
0 |
|
9 |
0.999 |
8.328 |
11.75 |
0 |
0 |
|
10 |
1.000 |
6.712 |
20.19 |
0 |
0 |
|
Abbreviations: CV: Coefficient of Variation. Dilution factor for all samples was 7.5. |
|||||
- In Line 108-109, authors wrote that increased AHI and decreased age were significantly associated with elevated 8-isoprostane levels. While the univariate analysis demonstrates this is statistically significant, the r correlation value is 0.16 and 0.11 respectively which indicates a very minor/weak association. Please rephrase for accuracy.
We agree and have edited the text to reflect this in a more accurate way. Lines 111-112 are now as follows: “Results of the univariate analyses are show in Table 2. In univariate analyses, AHI and age were significantly associated with elevated 8-isoprostane levels; however the correlation (r) was limited.”
Sincerely yours,
Bernardo Peres, Ismail Laher, and Najib Ayas on behalf of the authors
Round 2
Reviewer 2 Report
Authors have revised manuscript appropriately and addressed all my comments.